# Phylogeographic characterization of *Burkholderia pseudomallei* isolated from Bangladesh

Md. Shariful Alam Jilani[1]*, Saika Farook[1], Arittra Bhattacharjee[2], Lovely Barai[3], Chowdhury Rafiqul Ahsan[4], Jalaluddin Ashraful Haq[1], Apichai Tuanyok[5]

**1** Department of Microbiology, Ibrahim Medical College, Dhaka, Bangladesh, **2** Bioinformatics Division, National Institute of Biotechnology, Savar, Dhaka, Bangladesh, **3** Department of Microbiology, BIRDEM General Hospital, Dhaka, Bangladesh, **4** Department of Microbiology, University of Dhaka, Dhaka, Bangladesh, **5** Department of Infectious Diseases and Immunology, College of Veterinary Medicine, University of Florida, Gainesville, Florida, United States of America

* jilanimsa@gmail.com

**Data Availability Statement:** All data are in the manuscript and supporting information files.

**Funding:** The author(s) received no specific funding for this work.

## Abstract

### Background

*Burkholderia pseudomallei* possesses a diverse set of genes which encode a vast array of biological functions reflecting its clinical, ecological and phenotypic diversity. Strain variation is linked to geographic location as well as pattern of land uses. This soil-dwelling Gram-negative pathogen causes melioidosis, a tropical disease endemic in northern Australia and Southeast Asian regions including Bangladesh. Phylogeographic analyses of *B. pseudomallei* isolates by molecular typing techniques could be used to examine the diversity of this organism as well as to track melioidosis epidemics.

### Methods

In this study, 22 *B. pseudomallei* isolates, of which 20 clinical and two soil isolates were analyzed, utilizing Real-time PCR assay and multilocus sequence typing (MLST). The sequences were then submitted to PubMLST database for analysis and construction of phylogenetic tree.

### Findings

A total of 12 different sequence types (STs) that includes four novel STs were identified for the first time. Strains having STs 1005, 1007 and 56 were the most widespread STs frequently isolated in Bangladesh. ST 1005, ST 56, ST 1007 and ST 211 have been detected not only in Bangladesh but are also present in many Southeast Asian countries.

### Significance

ST 1005 was detected in both soil and clinical samples of Gazipur. Most prevalent, ST 56 has been previously reported from Myanmar, Thailand, Cambodia and Vietnam, confirming the persistence of the genotype over the entire continent. Further large-scale study is

**Competing interests:** The authors declare no conflict of interest.

necessary to find out the magnitude of the infection and its different reservoirs in the environment along with phylogeographic association.

## Author summary

Melioidosis, caused by *Burkholderia pseudomallei*, is a disease of various manifestations and poor outcome, if not diagnosed and treated early. The first case of melioidosis is believed to be identified from Bangladesh as early as 1960, although the persistence of the causative organism was discovered in soil about 50 years later, in 2012. The study utilized 20 *B. pseudomallei* isolated from clinical samples and 2 from soil samples. Multilocus sequence typing (MLST) was conducted with all 22 isolates and the sequence types analyzed and compared with *B. pseudomallei* sequence types previously detected from other countries. Four novel sequence types (STs) were detected in our study. Out of 12 different STs determined, STs 1005, 1007 and 56 were the commonest sequence types identified in Bangladesh. ST 1005 was found in both clinical and soil samples. ST 56 was isolated from five septicemic patients. The study revealed the sequence types of *B. pseudomallei* isolated from Bangladesh and their genetic relatedness with sequence types of other regions. We further showed that the sequence types of Bangladesh share more genetic similarities with the isolates of Southeast Asian isolates rather than South Asian isolates.

## Introduction

The soil-dwelling bacterium *Burkholderia pseudomallei* is responsible for causing the deadly disease melioidosis. Once considered an obscurity, the disease is now recognized as an emerging fulminating infectious disease of global significance. The hyper-endemic foci of this disease exist in northern Australia, Thailand and Singapore, where extensive surveillance had undertaken [1,2]. Since melioidosis is a disease of rural poor that usually affects the people in areas that are poorly supplied with the diagnostic capability to make the diagnosis, so the true burden of this disease throughout the globe remains abstruse [3]. However, in recent years melioidosis has been increasingly reported from diverse tropical regions of the globe where it has not been detected previously [4–6]. It is not clear whether the spread of melioidosis beyond the known endemic regions has occurred in recent decades, or it is just being unveiled by better recognition, increased awareness and surveillance. It has been hypothesized that *B. pseudomallei* may have originated in Australia and subsequently spread to South and Southeast Asia via animal migration during the Miocene period about 15 million years ago when the continents were joined by a land bridge due to greatly lowered sea levels [7]. From Asia *B. pseudomallei* appears to have spread to Africa and from there to the Americas, possibly via slave trading from West African Countries to America [8,9].

The genome of *B. pseudomallei* encompasses a diverse set of genes which encode a vast array of biological functions reflecting its clinical, ecological and phenotypic diversity. Strain variation within *B. pseudomallei* is linked to geographic location as well as pattern of land uses [10]. It has been determined that Australian isolates usually possess an ancestral *B. thailandensis*-like flagellum and chemotaxis (BTFC) gene cluster, whilst isolates from Asia and elsewhere almost exclusively carry a *Yersinia*-like fimbrial (YLF) gene cluster [11,12]. *B. pseudomallei* strains containing the YLF gene cluster are more associated with clinical cases, whereas strains containing the BTFC gene cluster are more likely to be environmental isolates, indicating a

potential role for the YLF gene associated with virulence [11]. The variation in clinical manifestations between melioidosis in Australia and Southeast Asia suggests that these populations are genetically distinct due to broad-scale bio-geographical factors associated with the establishment and persistence of the organism [11]. Melioidosis is caused by contact with contaminated surface water or soil, via respiratory route with human-to-human transmission being exceptionally rare [13]. Thus, large geographic barriers, as for example, the Wallace Line, which separates most of Asia from Australia, may have subsequently restricted gene flow between populations. However, whole-genome sequencing has resolved that Asian isolates of *B. pseudomallei* share an Australian ancestral root [14].

Melioidosis has been sporadically detected in Bangladesh over last several decades [15]. Until 2021, a total of 86 *B. pseudomallei* isolates have been obtained from 68 melioidosis cases [16]. Although, the first confirmed case of melioidosis from Bangladesh is believed to be identified in a 29-year-old British sailor in 1960, the persistence of the organism in soil was first confirmed 52 years later in 2012 [17,18]. In 2013, melioidosis endemic countries were categorized into "definite" and "probable" on the basis of the presence of the causative organism in human as well as environment [19]. *B. pseudomallei* was isolated for the first time in Indian Subcontinent from soil samples from Gazipur district of Bangladesh. Since then, Bangladesh is considered as a definitive country for melioidosis due to the presence of *B. pseudomallei* in both clinical and soil samples [18]. However, the true burden of the disease as well as mortality rates attributed to melioidosis are unknown, due to a lack of extensive epidemiological surveillance programs. Molecular analyses of *B. pseudomallei* isolates by multilocus sequence typing (MLST) can be used to examine the diversity of this organism from various geographic regions, which can assist with assessment of population structure of the organism as well as form phylogeographic characteristics that can be used to track melioidosis epidemics [20].

The current study demonstrates the utility of the MLST scheme for epidemiological studies and clarifies the genetic relationships between *B. pseudomallei* isolated from the clinical specimen and environmental sources. Understanding the population genetic structure and existing biodiversity of *B. pseudomallei* in endemic environments has implications for tracing outbreak source [21]. Detailed phylogenetic study of isolates will advance understanding of the biogeography of *B. pseudomallei* from this region as well as from the neighboring countries. In the present study, we examined the strains isolated from both clinical specimen and soil samples from different areas of Bangladesh using microbiological and molecular tests. To explore the genetic relatedness of our local isolates with isolates detected previously from other countries, we collected and compared sequences from PubMLST database [22].

## Methodology

### Ethics statement

The Ethical Review Committee (ERC) of the Diabetic Association of Bangladesh (BADAS) has approved the study (IMC/RP/2017/94). Ibrahim Medical College (IMC) is an institution under the BADAS, where the study was conducted.

### Sample collection and isolation of Bacterial strains

The study utilized a total of 23 *B. pseudomallei* isolates that included 20 clinical, two soil isolates and one reference strain (USM strain) obtained from Universiti Sans Malaysia. The study availed archived clinical isolates, which were obtained from diabetic patients who attended Bangladesh Institute of Research and Rehabilitation in Diabetes, Endocrine and Metabolic Disorders (BIRDEM General Hospital) from 2009 to 2015 and two soil strains isolated from Kapasia, Gazipur as previously described [15,18]. The sample collecting regions of Bangladesh

has been demonstrated in Fig 1 using DIVA-GIS (www.diva-gis.org). Bacteria were stored in Trypticase Soy Broth supplemented with 15% glycerol. Isolates were plated onto Ashdown's agar and cultivated at 37˚C for 48 hours before DNA extraction [23]. Real-time PCR assay followed by MLST from extracted DNA was performed at the Emerging Pathogens Institute, University of Florida, USA.

## Real-time PCR assay

From the extracted DNA Real-time PCR assay was done for molecular identification of Type III secretion system 1 (TTS1). Primer pair and probe were used to target the TTS1 gene cluster of *B. pseudomallei* as previously described [24]. The YLF and BTFC gene clusters were detected using Real-time PCR assay developed by Tuanyok A et al in 2007. PCR primers for detecting YLF and BTFC genes were used as previously described [11].

## Multilocus sequence typing (MLST) of the bacterial strains

MLST was performed by amplifying and sequencing of the seven housekeeping gene fragments (*ace*, *gltB*, *gmhD*, *lepA*, *lipA*, *narK* and *ndh*).The sequences were submitted to the PubMLST database (https://pubmlst.org/) for further analysis [22,25]. The presence of the Sequence Type (ST) in organisms detected in other countries was identified by exploring the PubMLST datasets for *B. pseudomallei*. Genetic relations between the isolates from Bangladesh and other countries that have found the same ST were visualized using GrapeTree [26].

## Characterization of the New ST

To characterize the genomic makeup of the new ST, the nucleotide sequences of the selected housekeeping genes were taken to construct a neighbor-joining (NJ) based phylogenetic tree with the *B. pseudomallei* sequence types isolated from Bangladesh [27]. The tree was constructed via PubMLST and visualized by iTOL [28]. After that the Based upon Related Sequences (BURST) algorithm was applied on the isolates of Bangladesh, Myanmar and Thailand to identify the relations of the genotypes in major clonal clusters with default group definition. The central ST of a BURST group have the highest number of single locus variants (SLVs). Generally, the most frequent ST take the central roles in the cluster/clonal complex and therefore helps to characterize the genotypic properties of the new ST [29,30].

# Result

## Isolates were collected from melioidosis patients and soil

Out of 20 patients from whom clinical samples were collected, 6 patients came from Gazipur while 5 patients were from Tangail. However, only one patient reported from Khagrachari which is geographically closer to Southeast Asian country Myanmar (Fig 1). Both isolates from soil samples were obtained from Gazipur [18].

## Real-time PCR identified all isolates as YLF positive *B. pseudomallei*

Real-time PCR assay, targeting TTS1, confirmed that all 20 clinical and two environmental isolates were *B. pseudomallei* along with the USM reference strain. All isolates contain YLF gene cluster and are negative to BTFC gene cluster.

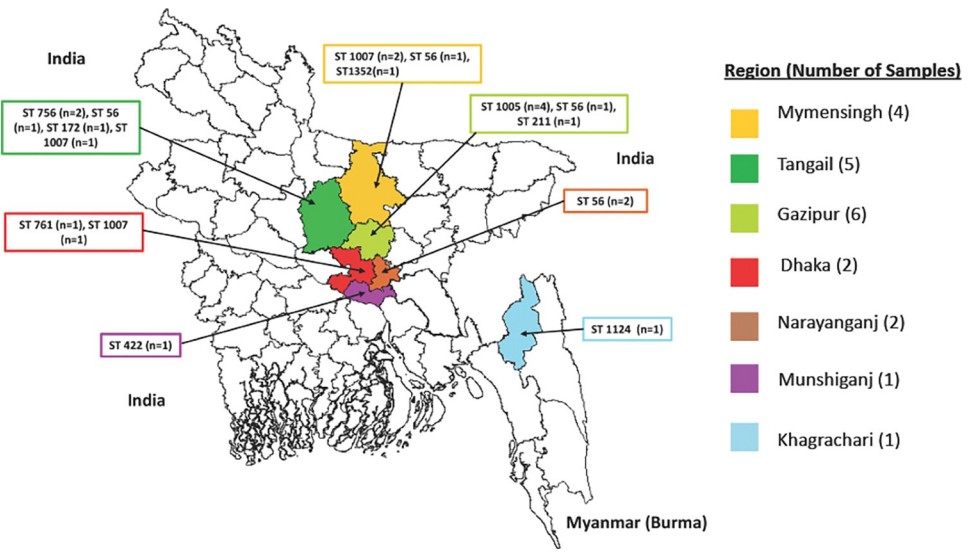

**Fig 1. Geographical locations of the melioidosis cases (www.diva-gis.org).**

## MLST profiling of *B. pseudomallei* from Bangladesh

MLST unveiled a total of 12 STs of which four STs (STs- 756, 1352, 761 and 61) were novel types and identified for the first time (Table 1). Strains having STs 1005, 1007 and 56 were the most frequently isolated STs in Bangladesh. ST 1005, ST 56, ST 1007 and ST 211 have been detected not only in Bangladesh but are also present in Thailand, Cambodia, Australia, Myanmar, China and Vietnam. According to PubMLST database, ST 1005 was previously isolated from soil samples of different regions such as environmental samples from Thailand and Australia. Interestingly, ST 1005 has not been isolated from clinical samples previously from any patients of any country. In Bangladesh, we have isolated two ST 1005 *B. pseudomallei* genotypes for the first time, from sputum and urine samples respectively. Whole genome sequencing study can provide further information about any possible linkage between the soil and clinical ST 1005 isolates.

**Table 1. Sequence Type (ST) of *B. pseudomallei* in the present study. ST that were common in other countries are also given.**

| Isolates reported from Bangladesh | | | No. of strains listed on PubMLST database | ST reported from Bangladesh and other geographic locations |
|---|---|---|---|---|
| ST | No. of isolates | Clinical | Environmental | | |
| 1005 | 4 | 2 | 2 | 25 | Thailand, Australia |
| 56 | 5 | 5 | 0 | 18 | Myanmar, Thailand, Cambodia & Vietnam |
| 756 | 2 | 2 | 0 | 2 | *Only in Bangladesh* |
| 1007 | 4 | 4 | 0 | 6 | Thailand |
| 211 | 1 | 1 | 0 | 14 | Thailand, China, Vietnam |
| 172 | 1 | 1 | 0 | 2 | Thailand |
| 422 | 1 | 1 | 0 | 17 | Singapore, Malaysia |
| 1352 | 1 | 1 | 0 | 1 | *Only in Bangladesh* |
| 761 | 1 | 1 | 0 | 1 | *Only in Bangladesh* |
| 1124 | 1 | 1 | 0 | 1 | India |
| 61 | 1 | 1 | 0 | 1 | *Only in Bangladesh* (Expatriate in Brunei) |

ST = Sequence Type

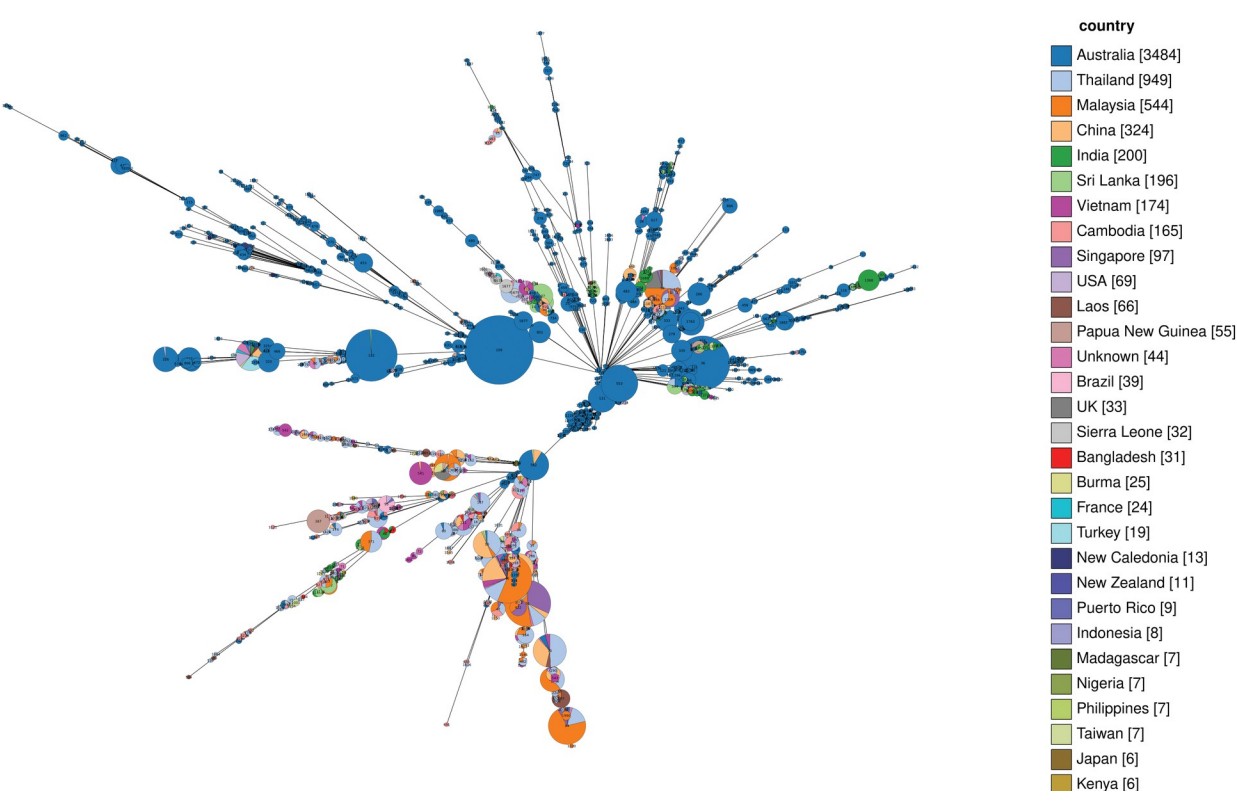

**Fig 2. Genetic relationships of *B. pseudomallei* isolates between the countries that share common sequence types with Bangladesh.**

## Predominant sequence types of Bangladesh share more genetic similarities with Southeast Asian strains rather than South Asian

Relationships between the Sequence Type were determined by GrapeTree (Fig 2). GrapeTree implemented a novel minimum spanning tree algorithm to rebuild these relationships [26]. According to the Grape Tree of PubMLST, two distinctive clusters were observed. In one cluster Australian strains such as ST 553, ST 131 were present, while in the other cluster, all strains were connected to Australian/ Asian strain ST 562. This ST 562 generated cluster is less predominant with Australian isolates and seems more diverse (Fig 2).

Microreact revealed close distance among the isolates of Myanmar, Thailand and Bangladesh via NJbased tree (S1 Fig) [31]. Two field breakdowns of the dataset revealed the frequencies and the mutual presence of ST 56 in Myanmar, Thailand and Bangladesh. Sequence Type 56 is predominant in both Bangladesh and Myanmar (Fig 2).

## Novel ST are closely related to Southeast Asian genotypes (especially Thailand)

Four novel genotypes (STs-1352, 756, 61 and 761) were found in this study. Among the sequence types of Bangladesh, ST 761 is clustered along with ST 1124 and ST 1548 (Fig 3). Sequence Type 1352 clustered with the clade of STs-1007 and 46. Sequence Type 756 did not cluster in any clades (Fig 3). Members of the same clades share more genetic similarities than distant clades. Sequence types- 1352, 756 and 761 were closely related with STs-1007, 188, 211, 874, respectively (S3 Fig). BURST analysis depicted that STs-1352 and 756 are double locus variant (DLV) of ST 48. Sequence Type 761 is a single locus variant (SLV) of ST 185 (S2 Fig).

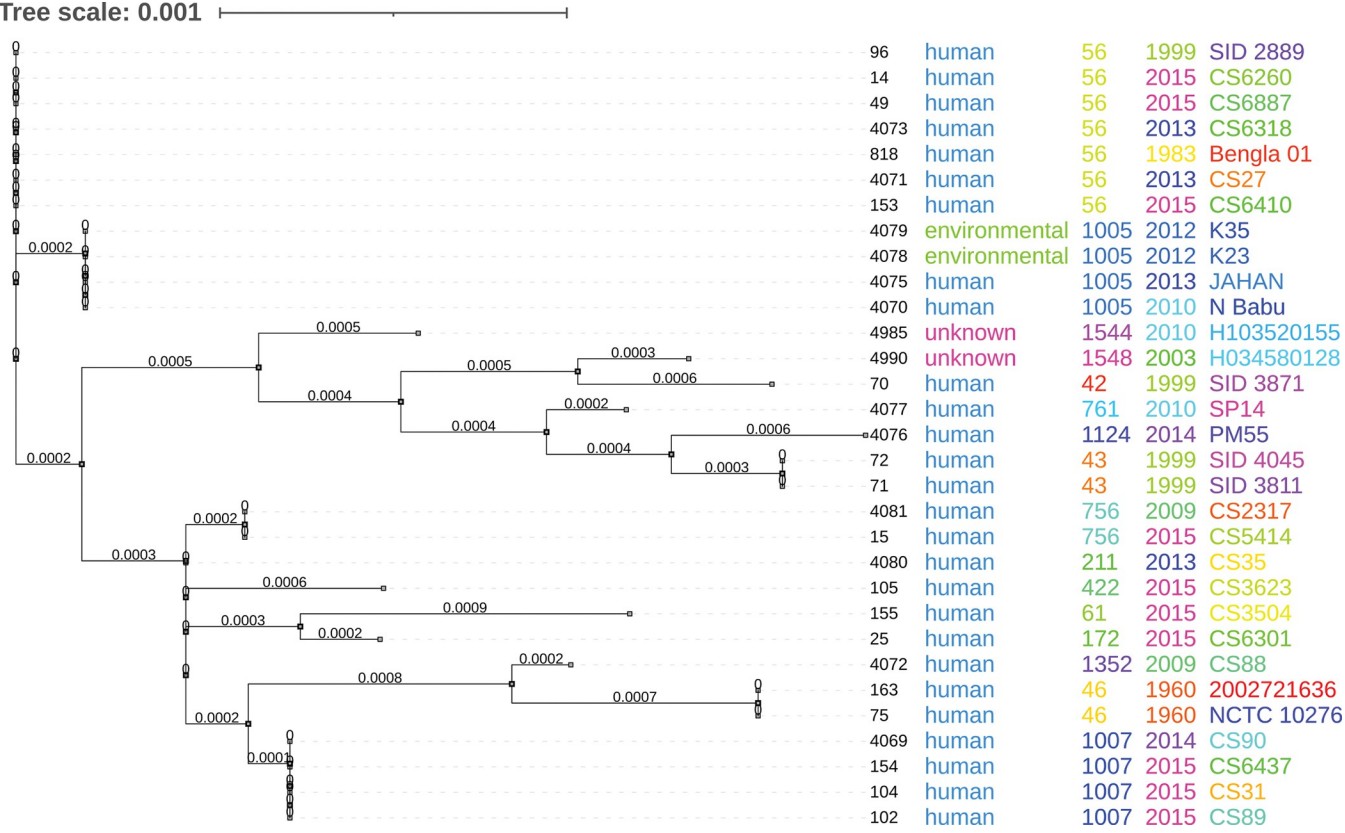

**Fig 3. Phylogenetic relationships of *B. pseudomallei* in Bangladesh.** On the right side, the columns depict the ID of the strain in the PubMLST database, Sequence Type (ST), specimen, source of the isolate and year of isolation.

All these central genotypes were found in Thailand or other Southeast Asian countries such as Vietnam and China.

## Discussion

Melioidosis, a severe disease caused by *B. pseudomallei*, is increasingly recognized in tropical and subtropical areas worldwide. In recent years, greater awareness among clinicians and microbiologists along with better laboratory facilities in Bangladesh aided in the isolation of the organism in growing numbers thus necessitating the understanding of population structure of the pathogen.

Real-time-PCR assay targeting both YLF and BTFC gene cluster in our clinical and soil isolates detected the presence of only YLF gene cluster in Bangladesh. YLF gene cluster is horizontally acquired while BTFC gene cluster is ancestral to *B. pseudomallei* [11]. The YLF group predominates in Asia while BTFC group is more prevalent in Australia. It is interesting to note that the YLF group is more common in clinical cases whereas BTFC group is more frequent in environmental isolates [11]. In Thailand, where YLF is dominant, mortality rates for the primary disease are higher (50%) than the endemic regions of Australia, where mortality rate is around 15 to 20% [32,33]; although high mortality rates are attributable to several confounding factors. Since the YLF gene is also dominant among the *B. pseudomallei* isolated in Bangladesh, further genomic studies may reveal whether virulence or disease severity is associated with the YLF gene that may lead to increase mortality rate.

The phylogenetic tree showed two groups of sequence types. One group belongs to ST 56 while another group (non-ST 56) has high sequence diversity (Fig 3). This sequence diversity indicates the endemicity of the pathogen [34,35]. Sequence types 1005, ST 56, ST 1007 and ST 211 have been detected not only in Bangladesh but are also present in Thailand, Cambodia, Australia, Myanmar, China and Vietnam. Additionally, *B. pseudomallei* STs 42, 43, 46, and 71 were previously identified from Bangladeshi patients [25]. Sequence Type 56 which was present in five clinical isolates is the most common variant present in Bangladesh, followed by ST 1005 and ST 1007.

According to PubMLST, at least four STs (STs-1352, 756, 761 and 61) were novel types and identified for the first time. It can be assumed that the existence of novel genotypes point towards local persistence of *B. pseudomallei* in the same geographical area and their ability to establish a new clone series producing novel offspring that carry new genotypes [36]. MLST based molecular surveillance underlined a strong genetic relationship among Chinese, Indian, Myanmar and Southeast Asian *B. pseudomallei* isolates [34–38]. All of these studies found common ST with that of strains isolated from Thailand where melioidosis is widespread. The genetic relationship of all the available ST from Bangladesh was visualized via GrapeTree. The Grape Tree clusters suggest that Australian strains such as ST 553, ST 131 present are distinctive from Australian/ Asian strain ST 562 which is related with the strains found in Bangladesh (Fig 2). Most plausibly, ST 562 went under different selection pressure and their variants were introduced in Bangladesh via tropical south-eastern part of Bangladesh (Myanmar and China). GrapeTree also reveals Sequence Type 56 are isolated from neighboring countries, Myanmar and Thailand. In addition, other sequence types like ST 172, ST 211 and ST 422 share more genetic similarities with Southeast Asian strains than South Asian isolates (Fig 2). Only, ST 1124 is found to be common between India and Bangladesh. This observed relatedness may be explained by *B. pseudomallei* transmission into Bangladesh due to geographical propinquity to neighbors, human and animal migration, ancient and persisting trading and travel networks. Previous studies demonstrated that Malay Peninsula was repeatedly reintroduced with *B. pseudomallei* [9,34]. However, to identify this type of reintroductions in Bangladesh, comparative genomics-based investigations are necessary.

The most common variant (i.e., ST 56) found in Bangladesh, was isolated from patients of diverse regions of the country. ST 56 was also isolated from a patient in 1983, as per PubMLST database, which suggests the existence of the genotype in the country for over a period of 32 years. ST 56 has been previously reported from Myanmar, Thailand, Cambodia and Vietnam, confirming the persistence of the strain, causing infection over the entire continent. The five ST 56 that were isolated in Bangladesh were from patients' blood (S1 Table). Two out of these five septicemic patients died. The first melioidosis case was detected in a Bangladeshi infant, suffering from pneumonia and septicemia, who died 12 hours following hospital admission [39]. This case was probably identified as the same ST 56 case in accordance with PubMLST data, which leads us to suspect that some distinct virulence factors are associated with this specific Sequence Type in Bangladesh. However, whole genome sequencing followed by in vivo and in vitro validations are required to provide clear pathophysiology.

The presence of *B. pseudomallei* is higher in anthrosol than acrisol soil types. Anthrosol is a soil type that has been highly modified by human for the preparation of paddy fields whereas acrisol is enriched with clay-rich soil common in tropical climates [40]. We found two ST 1005 isolates from anthrosol type of soil from Gazipur. ST 1005 was also isolated from clinical samples of Gazipur. These are the first clinical cases of ST 1005 according to PubMLST database. This strongly indicates that spatial clustering of clinical incidence is linked to the environmental persistence of the organism and infection follows environmental exposure. A study conducted by Roe et al (2022) analyzed sequence type 1005 isolated from soil. Following

inoculation of seven ST 1005 isolates in BALB/c mice, only one was found to be virulent, although it lacked the conserved gene (BPSS0417-BPSS0428) associated with virulence [41]. Analysis of the differences of virulent genes among ST 1005 isolated in different regions of the world will assist in understanding their ability to cause human infection.

The study included small number of isolates from sporadic melioidosis cases, indicating ecologically established population of *B. pseudomallei*. It contrasts with the study conducted in Australia by Mcrobb et al, which showed diversified population of the bacteria when increased sample size and sampling areas were considered [42]. Further large scale study will help to understand the true ecological establishment of *B. pseudomallei* in Bangladesh.

The detection of different sequence types in a defined area indicates the historical introduction and dissemination of different genotypes into the area or due to expansion of local ST that yielded new strains with novel ST [37]. MLST is an unambiguous and powerful tool to study the bacterial populations and its global distribution [25]. However, MLST is not very much efficient in detecting relatedness among *B. pseudomallei* ST due to high levels of lateral gene transfer in *B. pseudomallei* strain. In order to determine the origin of the organism in Bangladesh, its introduction and dissemination to other area and evolutionary change among strains require a whole genome-based single nucleotide polymorphism typing. Such study will also help us to understand the phylogeographic association of *B. pseudomallei* hailing from a particular region of the country with foreign strains and present a comparative analysis to identify essential genes involved in their pathogenicity, adaptability, and drug resistance mechanisms.

## Supporting information

**S1 Fig. Relationships between the strains of Bangladesh, Myanmar and Thailand.**
(TIF)

**S2 Fig. BURST (based upon related sequences) analysis of the novel STs. Novel STs are labeled with blue boxes**
(TIF)

**S3 Fig. Node that are Mostly close to the Novel STs (inside the red boxes).**
(TIF)

**S1 Table. Multi-Locus Sequence Typing Profile of the *B. pseudomallei* strains that have been isolated from Bangladesh.**
(XLSX)

## Acknowledgments

We thank Dr. Michael Norris, a former Postdoctoral Associate in Dr. A. Tuanyok's Laboratory, at the University of Florida for his technical assistance. We would also like to thank Dr. Mili Rani Saha, Dr. Tanzila Rahman and Md. Rokib Hasan, Department of Microbiology, BIRDEM Hospital for their assistance in detection, identification and preservation of the strains.

## Author Contributions

**Conceptualization:** Md. Shariful Alam Jilani, Chowdhury Rafiqul Ahsan, Jalaluddin Ashraful Haq.

**Data curation:** Md. Shariful Alam Jilani, Saika Farook, Arittra Bhattacharjee, Jalaluddin Ashraful Haq, Apichai Tuanyok.

**Formal analysis:** Md. Shariful Alam Jilani, Saika Farook, Arittra Bhattacharjee, Lovely Barai, Chowdhury Rafiqul Ahsan, Jalaluddin Ashraful Haq.

**Investigation:** Md. Shariful Alam Jilani, Jalaluddin Ashraful Haq, Apichai Tuanyok.

**Methodology:** Md. Shariful Alam Jilani, Saika Farook, Arittra Bhattacharjee, Chowdhury Rafiqul Ahsan, Jalaluddin Ashraful Haq.

**Project administration:** Md. Shariful Alam Jilani, Jalaluddin Ashraful Haq.

**Resources:** Md. Shariful Alam Jilani, Lovely Barai.

**Software:** Saika Farook, Arittra Bhattacharjee, Apichai Tuanyok.

**Supervision:** Md. Shariful Alam Jilani, Jalaluddin Ashraful Haq.

**Writing – original draft:** Md. Shariful Alam Jilani, Saika Farook, Arittra Bhattacharjee.

**Writing – review & editing:** Md. Shariful Alam Jilani, Saika Farook, Lovely Barai, Chowdhury Rafiqul Ahsan, Jalaluddin Ashraful Haq, Apichai Tuanyok.

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
