## [Decision Letter · Decision Letter 0]

3 Nov 2022

Dear Prof Jilani,

Thank you very much for submitting your manuscript "Phylogeographic characterization of Burkholderia pseudomallei isolated from Bangladesh" for consideration at PLOS Neglected Tropical Diseases. As with all papers reviewed by the journal, your manuscript was reviewed by members of the editorial board and by several independent reviewers. In light of the reviews (below this email), we would like to invite the resubmission of a significantly-revised version that takes into account the reviewers' comments. 

I am sorry about the delay to your manuscript, this was due to the need to find additional reviewers, which were required because the initial reviews did not converge on a common opinion. Consequently, we now have input from four reviewers, and the overall consensus is that a very major revision is required. Should you wish to resubmit a revised manuscript, please try to address all of the comments. In particular, please give particular emphasis to the comments of reviewers three and four, as these cover some critical points, and I think they are key to the relevance and validity of the study.

We cannot make any decision about publication until we have seen the revised manuscript and your response to the reviewers' comments. Your revised manuscript is also likely to be sent to reviewers for further evaluation.

Sincerely,

Stephen W. Attwood, BSc,MSc,PhD

Academic Editor

Elsio Wunder Jr

Section Editor

I am sorry about the delay to your manuscript, this was due to the need to find additional reviewers, which were required because the initial reviews did not converge on a common opinion. Consequently, we now have input from four reviewers, and the overall consensus is that a very major revision is required. Should you wish to resubmit a revised manuscript, please try to address all of the comments. In particular, please give particular emphasis to the comments of reviewers three and four, as these cover some critical points, and I think they are key to the relevance and validity of the study.

Reviewer's Responses to Questions

**Key Review Criteria Required for Acceptance?**

**Methods**

-Are the objectives of the study clearly articulated with a clear testable hypothesis stated?

-Is the study design appropriate to address the stated objectives?

-Is the population clearly described and appropriate for the hypothesis being tested?

-Is the sample size sufficient to ensure adequate power to address the hypothesis being tested?

-Were correct statistical analysis used to support conclusions?

-Are there concerns about ethical or regulatory requirements being met?

Reviewer #1: -The study objectives are clearly sated but there is no testable hypothesis stated

-There is no study design mentioned

-population not quite described and no appropriate sample size determined

-The study lacked rigorous scientific sampling procedure and spatial spread of sampling to infer phylogeographic distribution of the isolates.

- Data analysis were done using phylogenetic software domiciled at PubMLST but the analysis lacked the geographic aspect

- If there are evidence of institutional Ethical Committee approval it is adequate

Reviewer #2: The authors have not provided the accession numbers of the sequences submitted to PubMLST. Additionally application numbers for ethics have not been provided.

Apart from the above, the methods are sound and appropriate to address the study aims of investigating the dispersal of sequence types in Bangladesh.

Reviewer #3: The number of isolates used in this study is very limited (24 isolates). All the clinical isolates were obtained from DM patients in BIRDEM General Hospital from 2009 to 2015. Only 2 isolates were obtained from the environment of the same place and the same year. 

Most of the patients were reported to come from Gazipur and Tangail districts and 2 soil isolates also from Gazipur. 

Moreover, the sources of the isolates may not fit the title “phylogeographic characterization” even though the patients who attended that hospital came from different regions but near by the hospital. This also created a bias of ST types obtained in this study.

Reviewer #4: -Are the objectives of the study clearly articulated with a clear testable hypothesis stated? Somewhat

-Is the study design appropriate to address the stated objectives? Mostly

-Is the population clearly described and appropriate for the hypothesis being tested? Yes

-Is the sample size sufficient to ensure adequate power to address the hypothesis being tested? NA

-Were correct statistical analysis used to support conclusions? Yes

-Are there concerns about ethical or regulatory requirements being met? No

**Results**

-Does the analysis presented match the analysis plan?

-Are the results clearly and completely presented?

-Are the figures (Tables, Images) of sufficient quality for clarity?

Reviewer #1: -There was phylogenetic analysis but not much about the geographic aspect rather than few comparisons made

-Result presentations not quite clear as some important figures mentioned were cited to be in supplementary materials

Reviewer #2: The figures are appropriate but are of very low quality. This could be due to the PLOS system mangling the figures. Please upload high quality figures.

Reviewer #3: The MLST typing from 7 housekeeping genes sequences is a very useful molecular tool to subtype the organism and compared them with isolates from around the world. However, interpretation of the results and conclusion of isolates with the same ST in this study may not appropriate and sometime over claimed. 

For example, “ST 1005 was detected in both soil and clinical samples of Gazipur which indicates the soil to human transmission of this pathogen”. It is known that the bacterium can infect both human and animal with several evidences and this conclusion is not appropriate.

Lines 156: “However, according to PubMLST database we found, for the first time, two isolates with ST 1005 from clinical samples of humans residing in the same district”. This sentense doesn’t make any sense.

Lines 241-243: “Only, ST 1124 and ST 300 were common between India and Bangladesh. Therefore, most plausibly, the major introduction of B. pseudomallei took place via Southeast Asian regions in Bangladesh.” This may be over claimed.

Reviewer #4: -Does the analysis presented match the analysis plan? Mostly

-Are the results clearly and completely presented? Mostly

-Are the figures (Tables, Images) of sufficient quality for clarity? No. Figures are of poor quality

**Conclusions**

-Are the conclusions supported by the data presented?

-Are the limitations of analysis clearly described?

-Do the authors discuss how these data can be helpful to advance our understanding of the topic under study?

-Is public health relevance addressed?

Reviewer #1: -The data data may not be adequate as no sample size determined as well as no scientific sampling procedure employed for the data to support scientifically supported conclusions

-Limitations are not clearly stated however authors have as part of concluding part mentioned in line 294 " whole genome-based single nucleotide polymorphism typing"

-There are statements of public health significance of the organism in the write up but silent on the public health significance of the study findings

Reviewer #2: (No Response)

Reviewer #3: Discussion and conclusion needs more attention. 

for example, lines 213-218: In Thailand, where YLF is dominant, mortality rates for the primary disease are higher (50%) than the endemic regions of Australia, where mortality rate is around 15 to 20% [29,30]. In an earlier study conducted in Bangladesh, sero-positivity rate among healthy individual was found to be 22·6%-30·8% [15]. If reactivation occurs in these cases due to an immune-compromised state, the YLF dominance may play a vital role in elevating the mortality rates.”

The mortality rate depends on several factors in each country that infecting with the bacterium that carrying YLF genotype cannot indicate the outcome. Moreover, the idea if the reactivation occurred and extrapolate it to link with mortality rate is not suitable.

About ST 56 that commonly found in Bangladesh, the author mention in line 250 that ST 56 was also isolated from a patient in 1983. This requires a reference. Moreover, (lines 253-254) the presence of this ST type in the countries nearby is not enough to indicate that B. pseudomallei may spread from these countries to Bangladesh. Lines 256-257 concluded that ST 56 is virulence as all isolates with this ST type were isolated from patients’ blood and 2 out of 5 died. These interpretation and conclusion are not scientifically sound.

Reviewer #4: -Are the conclusions supported by the data presented? Not all. The authors over-interpret the significance of their findings a bit

-Are the limitations of analysis clearly described? No

-Do the authors discuss how these data can be helpful to advance our understanding of the topic under study? Yes

-Is public health relevance addressed? Yes

**Editorial and Data Presentation Modifications?**

Reviewer #1: (No Response)

Reviewer #2: Figure 2 is very chaotic and requires reworking to clearly show where the Bangladesh Bp samples sit.

Reviewer #3: Figure 2 and 3 needs more details for the interpretation and further clarify.

The information about each isolate were listed together with phylogenic tree in figure 4 that quite difficult to read.

Reviewer #4: (No Response)

**Summary and General Comments**

Reviewer #1: The Authors attempted to phylogenetic and geographic (phylogeographic) characterization of Burkholderia pseudomallei isolated from Bangladesh. Twenty four (24) isolates comprising of 22 clinical and two soil isolates were used in this study where they found a total of 13 different Sequence Type (ST) that includes five novel STs. The study lacked rigorous scientific sampling procedure and spatial spread of sampling to infer phylogeographic distribution of the isolates.

Reviewer #2: Overall the manuscript is well written, communicating clearly the methods, experimental design and conclusions. It provides important data on the geographical distribution of B. psueodmallei STs in Bangladesh. I recommend some improvements to the figures, including better quality and reworking figure 2.

Reviewer #3: The information about ST type of B. pseudomallei in Bangladesh may be novel, however, the number of isolate is limitted and come from only one hospital. The conclusion and interpretation throughout the manuscript is not scientific or reasonable that needs correction. The MLST and phylogenic analysis were done by a colleague in the USA that need more details clarification to be understandable by general readers.

Reviewer #4: First review of PNTD-D-22-00909

The authors make an important contribution with this submission by genotyping clinical and environmental B. pseudomallei isolates from Bangladesh. The manuscript can be improved by minor editing, improved figures, and by not overinterpreting their data.

Abstract – first line of significance section: Change “indicates” to “suggests”. This is a correlation, not causation

Abstract – significance section, Discussion – lines 251-254: The use of the word “strain” is problematic as it can mean very different things. A particular ST is not a “strain”. It is a genotype that may reflect a phylogenetic lineage.

Introduction – lines 38-39: Change “…a disease of rural poor and usually affects the people in the area that are poorly supplied with the diagnostic capability to make the diagnosis, so the…” to “…a disease of rural poor that usually affects people in areas that are poorly supplied with the diagnostic capability to make a diagnosis, the…”

Introduction – lines 44-47: Citations 5 and 8 are book chapters that summarize the primary literature. Better to not use these citations but, instead, go back to the original papers that made these points and were cited in these book chapters. For example, this point was first made by Pearson et al 2009, your citation #11. Credit for that observation should be given to that first paper, not the currently cited book chapter.

Introduction – lines 47-49: Change “From Asia B. pseudomallei eventually migrated to Madagascar and Africa and then through slave trading from West African Countries to America….” to ““From Asia B. pseudomallei appears to have spread to Africa and from there to the Americas, possibly via slave trading from West Africa””. Correct citations for this statement are your current citations #7 and #28, not your citation #6.

Introduction – line 52: Change “among” to “within”

Introduction – line 54: Change “from Asia” to “from Asia and elsewhere”

Introduction – line 56: Change “associated” to “more associated”

Introduction – line 57: Change “clusters” to “cluster”

Introduction – lines 55-58: Again, use the primary literature for this citation, not a book chapter. The proper citation for this statement is your current citation #9.

Introduction – line 58: Change “This” to “The”

Introduction – lines 59-61: No citation is listed to support this statement: “suggests that these populations are genetically distinct due to broad-scale bio-geographical factors associated with the establishment and persistence of the organism”. The point of local adapatation facilitated by local acquisition of new genomic regions was made in PLoS NTD 13:e000727

Introduction – line 61: Melioidosis is also obtained via the aerosol route

Introduction – line 62: Need a citation to support the statement that human to human transmission is rare. Bart Currie has made this point in some of his papers

Introduction – line 63: Change “…Australia, have…” to “…Australia, may have…”

Introduction – line 66: Delete the word “recent”. That paper is almost 15 years old.

Introduction – lines 68-70: Change “Till 2021, a total of 86 culture confirmed B. pseudomallei has been isolated from 68 melioidosis cases [13]. Although, the first case of melioidosis from Bangladesh is believed to be identified” to “Since 2021, a total of 86 B. pseudomallei isolates have been obtained from 68 melioidosis cases [13]. Although, the first confirmed case of melioidosis from Bangladesh is believed to have been identified”

Introduction – line 74: What is meant by “definitive country”?

Introduction – lines 75-79: Change “…to melioidosis is unknown, due to a lack of extensive epidemiological surveillance program. For that reason, molecular analyses of B. pseudomallei isolates by multilocus sequence typing (MLST) may be used to examine the diversity of this organism from various geographic regions. This will assist to assess population structure of the organism as well as form phylogeographic characteristics that can be…” to “…to melioidosis are unknown, due to a lack of extensive epidemiological surveillance programs. Molecular analyses of B. pseudomallei isolates by multilocus sequence typing (MLST) can be used to examine the diversity of this organism from various geographic regions, which can assist with assessment of population structure of the organism as well as be…”

Methods: Why weren’t the isolates whole genome sequenced?

Methods - line 118: Insert the citation for Godoy.

Methods – lines 127-128: What does “with the ST of Bangladesh” mean?

Methods – line 133: What does “genotypic behaviors” mean?

Results – lines 137-138: This sentence is redundant with text in the methods section.

Discussion – lines 210-211: YLF was acquired but BTFC is the ancestral state. Please rephrase.

Discussion – lines 213-215: The differences in mortalities rates between Thailand and Australia are largely attributed to differences in healthcare quality, not the YLF/BTFC difference.

Discussion – lines 215-218: Reactivation almost certainly would not occur in someone who has seroconverted.

Discussion – line 218: There is no evidence that YLF increases mortality of virulence. Please delete “the YLF dominance may play a vital role in elevating the mortality rates”.

Discussion – lines 239-243: I would be careful speculating that B. pseudomallei was introcuded to Bangladesh because there is more diversity in Thailand (i.e., more STs found there). There has been a lot more environmental/clinical sampling of B. pseudomallei in Thailand than anywhere else in Asia. More sampling will reveal more diversity as rarer types will be discovered. There may be just as much diversity, or more, in Bangladesh but it has not been revealed due to limited sampling.

Discussion – lines 259-261: There is nothing in this study to document that ST56 is more virulent. There may just be more clinical cases caused by it because it is so common.

Discussion – lines 272-274: What is the evidence to support this statement: “In addition, this may also point out that naturally occurring groundwater functions as a vehicle for the dispersal of viable B. pseudomallei away from a primary environmental reservoir”?

Figure 2: This figure is almost impossible to interpret.

Figure 3: The map portion of this figure is confusing and unneeded.

Figure 4: What are the values along the branches? What are the units of the scale bar?

PLOS authors have the option to publish the peer review history of their article (what does this mean?). If published, this will include your full peer review and any attached files.

Reviewer #1: No

Reviewer #2: No

Reviewer #3: No

Reviewer #4: No
---

## [Decision Letter · Decision Letter 1]

18 Oct 2023

Dear Prof Jilani,

Thank you very much for submitting your revised manuscript "Phylogeographic characterization of Burkholderia pseudomallei isolated from Bangladesh" for consideration at PLOS Neglected Tropical Diseases. Thank you also for your patience during this review, as many potential reviewers were on vacation during the northern summer, or otherwise engaged. As with all papers reviewed by the journal, your manuscript was reviewed by members of the editorial board and by several independent reviewers. The reviewers appreciated the attention to an important topic. Based on the reviews, we are likely to accept this manuscript for publication, providing that you modify the manuscript according to the review recommendations. 

There remain some minor corrections (i.e. they require only changes to the text), but important points to address. In particular, the points made by Reviewer 3 regarding the claims made; these must correspond to, and not go beyond, the analyses and results presented here. For example, the routes of entry to the country can only be stated as covered by the study if some phylogeographical or migration rate analysis had been done. Such analyses were not part of this study. There are other similar points that need to be addressed.

Sincerely,

Stephen W. Attwood, BSc,MSc,PhD

Academic Editor

Elsio Wunder Jr

Section Editor

There remain some minor corrections (i.e. they require only changes to the text), but important points to address. In particular, the points made by Reviewer 3 regarding the claims made; these must correspond to, and not go beyond, the analyses and results presented here. For example, the routes of entry to the country can only be stated as covered by the study if some phylogeographical or migration rate analysis had been done. Such analyses were not part of this study. There are other similar points that need to be addressed.

Reviewer's Responses to Questions

**Key Review Criteria Required for Acceptance?**

**Methods**

-Are the objectives of the study clearly articulated with a clear testable hypothesis stated?

-Is the study design appropriate to address the stated objectives?

-Is the population clearly described and appropriate for the hypothesis being tested?

-Is the sample size sufficient to ensure adequate power to address the hypothesis being tested?

-Were correct statistical analysis used to support conclusions?

-Are there concerns about ethical or regulatory requirements being met?

Reviewer #2: (No Response)

Reviewer #3: The objectives are clear but may not interpret and analyze well by a small number of isolates used in the study. This limitation should be stated somewhere.

**Results**

-Does the analysis presented match the analysis plan?

-Are the results clearly and completely presented?

-Are the figures (Tables, Images) of sufficient quality for clarity?

Reviewer #2: (No Response)

Reviewer #3: Quite low quality immages.

**Conclusions**

-Are the conclusions supported by the data presented?

-Are the limitations of analysis clearly described?

-Do the authors discuss how these data can be helpful to advance our understanding of the topic under study?

-Is public health relevance addressed?

Reviewer #2: (No Response)

Reviewer #3: Some points still overclaimed especially in the author summary.

**Editorial and Data Presentation Modifications?**

Reviewer #2: (No Response)

Reviewer #3: Minor revision including some typo error correction.

**Summary and General Comments**

Reviewer #2: (No Response)

Reviewer #3: Revised MS PNTD-D-22-00909_R1

The revised manuscript title “Phylogeographic characterization of Burkholderia pseudomallei isolated from Bangladesh.” was mostly corrected and changed according to what reviewers’ suggestion.

There are a few more comments to be concerned.

As McRobb, E. et al, Applied and Environmental Microbiol, 2014 indicated when analyzed B. pseudomallei populations in Australia (779 clinical isolates, including 736 humans plus 43 animal isolates, and 174 environmental strains) using ST analysis that B. pseudomallei populations diversified as the sampling area increased. This observation contrasted with smaller sampling areas where a few STs predominated, suggesting that B. pseudomallei populations are ecologically established and not frequently dispersed. Therefore, the interpretation using a small number of isolates, even though sending from surrounding areas, in this study should mentioned for their limitation.

This manuscript mentioned about ST1005 that isolated from 2 clinical samples and this ST has not been isolated from clinical samples previously from any patients of any country. Roe, C. et al, PLOS Neglected Tropical Diseases| 2022, analyzed ST1005 including the information that there is one virulent isolate, and six attenuated isolates were from the same ST (ST1005). Therefore, suggesting that variably conserved genomic regions may contribute to virulence. The author may discuss and add some information and cite this as a reference.

Minor points

1. The author summary may still over claim in some points such as: 

a. “Molecular typing studies may be utilized to understand the route through which this organism has entered the country”. This may be over elaborated. 

b. “The current study has demonstrated the utility of the multilocus sequence typing scheme for epidemiological studies and clarifies the genetic relationships between B. pseudomallei isolated from the clinical specimen and soil samples”. Only 2 soil isolates were analyzed and only one has similar ST with a clinical isolate.

c. “The findings that the same sequence type present in soil and clinical samples from the same region, will aid in future to track melioidosis outbreaks resulting in transmission breakdown with continued epidemiological monitoring. The study will also play a vital role in establishing a guideline for the prevention and treatment of melioidosis in Bangladesh as well as the basis of vaccine development.” This summary may too far beyond the outcome of this study.

2. Please check some citation for example: 

a. “Strain variation within B. pseudomallei is linked to geographic location as well as pattern of land uses [10].” The cited paper compared genes up- or down-expressed in vivo compared to cultures across the three genomes of Bm, Bp, and Bt.

b. “The variation in clinical manifestations between melioidosis in Australia and Southeast Asia suggests that these populations are genetically distinct due to broad-scale bio-geographical factors associated with the establishment and persistence of the organism [13]” The cited paper reported B. pseudomallei in the environment in Puerto Rico.

3. Please clarify the last sentence in the introduction. “We then compared the MLST data with those of the B. pseudomallei MLST database to determine the genetic relatedness and attempted to provide an insight into the ecology of B. pseudomallei through an elucidation of the mechanisms involved in evolution, speciation, and dispersal.” 

4. Lines 148-150: “The YLF and BTFC is a multiplex real-time PCR assay using SYBR green as a fluorescent dye targeting YLF and BTFC gene cluster ….” As YLF and BTFC are gene clusters, mentioned them as a multiplex real-time PCR assay may not appropriate.

5. Abstract line 37: put space between two words “pseudomalleiisolates”

PLOS authors have the option to publish the peer review history of their article (what does this mean?). If published, this will include your full peer review and any attached files.

Reviewer #2: No

Reviewer #3: No

Figure Files:

Data Requirements:

Reproducibility:

References

---

## [Editor Report · Decision Letter 2]

27 Nov 2023

Dear Prof Jilani,

We are pleased to inform you that your manuscript 'Phylogeographic characterization of Burkholderia pseudomallei isolated from Bangladesh' has been provisionally accepted for publication in PLOS Neglected Tropical Diseases.

Best regards,

Stephen W. Attwood, BSc,MSc,PhD

Academic Editor

Elsio Wunder Jr

Section Editor

---

## [Editor Report · Acceptance letter]

4 Dec 2023

Dear Prof Jilani,

We are delighted to inform you that your manuscript, "Phylogeographic characterization of Burkholderia pseudomallei isolated from Bangladesh," has been formally accepted for publication in PLOS Neglected Tropical Diseases.

Best regards,

Shaden Kamhawi

co-Editor-in-Chief

Paul Brindley

co-Editor-in-Chief
